Preparation and analysis of quinoa active protein (QAP) and its mechanism of inhibiting Candida albicans from a transcriptome perspective

Zhang Xufei
Zheng Chunmei
Ge Wenxuan
Li Xueying
Wang Xiuzhang
Sun Yanxia sunyanxia1976@cdu.edu.cn
Wu Xiaoyong cduwxysyx@126.com
Key Laboratory of Coarse Cereal Processing, Ministry of Agriculture and Rural Affairs, Sichuan Engineering and Technology Research Center of Coarse Cereal Industrialization, College of Food and Biological Engineering, Chengdu University , Chengdu , Sichuan Province , China
Gawande Suresh
Electronic publication date: 2025 Feb 14
Publication date: 2025
Volume: 13
Electronic Location ID: e18961
Received 2024 Oct 28; Accepted 2025 Jan 20
Copyright: ©2025 Zhang et al.
Copyright year: 2025
Copyright holder: Zhang et al.
License: This is an open access article distributed under the terms of the Creative Commons Attribution License, which permits unrestricted use, distribution, reproduction and adaptation in any medium and for any purpose provided that it is properly attributed. For attribution, the original author(s), title, publication source (PeerJ) and either DOI or URL of the article must be cited.
License URL: https://creativecommons.org/licenses/by/4.0/

Keywords: Chenopodium quinoa, QAP, C. albicans, Proteomic, Transcriptomics

Funding: Project of Sichuan Provincial Administration of Traditional Chinese Medicine 2023MS273 This research was funded by the Project of Sichuan Provincial Administration of Traditional Chinese Medicine (2023MS273). The funders had no role in study design, data collection and analysis, decision to publish, or preparation of the manuscript.

==============================
The globe cultivation and industrial production of quinoa have been steadily increasing. Nevertheless, the full potential of quinoa’s nutritional and economic benefits remains underexploited. This study investigates the isolation and purification of quinoa active protein (QAP) through the precipitation method using ammonium sulfate from a phosphate extraction solution. The purification process involved gel filtration chromatography with Sephadex G-75 and Sephadex G-50 columns to obtain QAP fractions exhibiting inhibitory effects against Candida albicans (C. albicans). A comprehensive series of experiments was undertaken to examine the antifungal properties of these fractions. Proteomic analysis was employed to elucidate the composition of the active proteins. Furthermore, the activities of succinate dehydrogenase, Ca2+-Mg2+-ATPase, and catalase in C. albicans following treatment with QAP were quantified using an enzyme-linked immunosorbent assay. The effects of QAP on the cell morphology of C. albicans cultured on Spider agar medium was further investigated using scanning electron microscopy (SEM). Furthermore, RNA-seq analysis was conducted to investigate the alterations in gene expression in C. albicans cells subjected to QAP treatment. To elucidate the functional significance of these expression changes, Gene Ontology and Kyoto Encyclopedia of Genes and Genomes analyses were performed. Quantitative real-time polymerase chain reaction was subsequently employed to validate the observed changes in gene expression. Our findings demonstrate that QAP exhibits inhibitory effects against C. albicans, with a minimum inhibitory concentration of 182 µg/mL. Through proteomic analysis, a definitive set of 18 active quinoa proteins was identified. At the molecular level, genes associated with hyphal development, cell wall and membrane integrity, cellular respiration, and energy metabolism were found to be enriched. Protein–protein interaction (PPI) analysis revealed that these QAP inhibit the growth of C. albicans hyphae, compromise cell wall and membrane integrity, and suppress oxidative phosphorylation. These disruptions collectively impair normal cellular metabolic activities, thereby exerting an inhibitory effect on C. albicans.

Introduction

Quinoa (Chenopodium quinoa Willd.), an indigenous plant of South America and a member of the Chenopodiaceae family, has attracted significant interest from the agricultural sector (Carciochi et al., 2016) owing to its relatively elevated starch content, ranging from 53.2 to 75.1 g/100 g (Wu, Morris & Murphy, 2017), as well as its high protein content, which includes all nine essential amino acids crucial for various physiological functions (Agarwal et al., 2023). This crop is noted for its substantial nutritional value and its adaptability to a wide range of ecological and climatic conditions (Habib, Ijaz & Haq, 2023). Consequently, several countries, including China, India, and Canada, have commenced large-scale cultivation of quinoa. Quinoa seeds, predominantly characterized by their round shape and white color, are also available in red and dark black varieties (Nowak, Du & Charrondière, 2016). In compared to conventional grains such as wheat, quinoa is notable for its enhanced nutritional profile, distinct starch properties, and higher dietary fiber content (Mu et al., 2023). Additionally, quinoa is recognized as an exceptional source of essential minerals and vitamins, and it is abundant in bioactive compounds like polyphenols and flavonoids, which are associated with numerous health benefits (Abugoch James, 2009). In recent years, significant research endeavors have focused on exploring the potential of quinoa as a functional food, aiming to maximize the utilization of this valuable resource (Jan et al., 2023).

Furthermore, the abundant presence of bioactive compounds in quinoa underscores its potential candidate for biomedical applications, particularly in the prevention of chronic diseases (Vega-Gálvez et al., 2010). Among these bioactive compounds, quinoa proteins demonstrate considerable developmental potential (Barakat et al., 2024). Vilcacundo, Martínez-Villaluenga & Hernández-Ledesma (2017) utilized proteomics techniques to analyze the components of quinoa protein and discovered that the peptides derived from 11S seed storage globulin B possess potential anti-diabetic activity. As a fundamental nutrient essential for human physiological functions, protein is integral to maintaining health and well-being (Guo et al., 2021). Due to its abundant protein content (Rubinovich et al., 2023) and balanced amino acid composition (Galindo-Luján et al., 2023), quinoa exhibits high biological value (Aprodu & Banu, 2021), and progressively gained recognition as a valuable source of plant-based protein within the food industry (Gu et al., 2023). Among the various parts of the plant, mature quinoa seeds, as the storage organs for nutrients, accumulate most of the protein in quinoa. The protein composition of mature quinoa seeds is predominantly composed of globulins, which constitute approximately 37% of the total protein, followed by albumins, which are present in lower concentrations (0.5–7% of the total protein) (Dakhili et al., 2019). Currently, prevalent methods for the extraction of quinoa protein encompass alkali extraction and acid precipitation techniques (Cui et al., 2023), as well as the salting-out method (Liu et al., 2023).

In natural ecosystems, higher plants generally coexist with a diverse range of microorganisms and pests within a common environment during various growth stages, including seed storage and plant establishment. Over the course of long-term evolution, higher plants have developed intricate defense mechanisms to counteract the invasions of deleterious organisms, including bacteria and fungi (Iqbal et al., 2019; Li et al., 2021). For instance, proteins with antimicrobial properties primarily include chitinase and β-1, 3-glucanase (Yang et al., 2007). These non-specific proteins, collectively referred to as antimicrobial proteins, are essential components of the plant’s innate immune defense system. They are characterized by a broad antimicrobial spectrum, strong antimicrobial activity, and the advantage of not inducing drug resistance.

Furthermore, antimicrobial proteins are predominantly found in plant seeds (Farvardin et al., 2024). It has been documented that proteins isolated from Medicago sativa seeds exert an inhibitory effects on Verticillium dahlia (Fradin & Thomma, 2006), while proteins extracted from Momordica charantia seeds demonstrate antifungal activity against species such as Aspergillus niger (Zhu et al., 2020). Proteins derived from Moringa seeds have been demonstrated to significantly inhibit the growth of Bacillus pumilus, as well as other microorganisms (Soliman, Arab & Soliman, 2021). However, there remains a paucity of research regarding the antimicrobial properties of proteins derived from quinoa.

Candida albicans (C. albicans) is a common symbiotic fungus (Jha & Kumar, 2024) and an opportunistic pathogenic fungus (Alkhars et al., 2024; Kumar & Kumar, 2023), recognized as the primary etiological agent of human fungal infections. Although C. albicans typically does not affect healthy individuals, it can readily cause superficial mucosal infections (Li et al., 2016) or fatal systemic infections (Li et al., 2015) in immunocompromised individuals (Mahmoud et al., 2022), leading to a high mortality rate (Carrano et al., 2024). Annually, more than 6.5 million individuals globally contract infections caused by invasive fungal infections, resulting in up to 3.75 million fatalities (Denning, 2024) and substantial economic repercussions (Williams, Cleary & Thomas, 2023). Among them, the infections caused by C. albicans account for 40–60% of all invasive fungal infections (Bays et al., 2024). A further significant concern for human health is the propensity of C. albicans to adhere to diverse medical device surfaces (Lee et al., 2016) and form biofilms (Biniarz et al., 2015), which markedly contributes to the incidence of invasive candidiasis (Sharma, Parmanu & Sharma, 2024). The process of filamentous growth is integral to biofilm formation, as it necessitates the expression of multiple filament-associated genes essential for surface adhesion (Woolford et al., 2016). C. albicans exhibits the ability to transition from its oval yeast form to a filamentous mycelium form. This morphological plasticity is intricately linked to its adaptability and pathogenicity (Arkowitz & Bassilana, 2019), thereby presenting significant challenges for the clinical management of candidiasis resulting from C. albicans infection (Lee et al., 2022). Despite the availability of clinical drugs for the treatment of C. albicans, which target various mechanisms—such as echinocandins (disrupting β-1,3-glucan synthesis in cell walls) (Dantas et al., 2021), fluconazole (FLC, inhibiting ergosterol synthesis) (Jahanshiri et al., 2022), and other antifungal medications (Maraki et al., 2015), the emergence of drug resistance (Meccatti et al., 2022) and the prevalence of side effects have become increasingly prominent concerns (Kniemeyer et al., 2018). Consequently, this issue has escalated into a significant global concern necessitating immediate intervention (Awassa et al., 2024). As a result, the pursuit of innovative anti-C. albicans compounds sourced from natural products has emerged as a central objective in contemporary drug development endeavors (Ye et al., 2023).

Therefore, the aim of this study was to isolate the quinoa active protein (QAP) with inhibitory properties against C. albicans from quinoa seeds. To achieve this, the highly sensitive and nearly 100% reliable LC-MS/MS proteomics approach, along with the advanced AB SCIEX Triple TOF™ 5600 plus mass spectrometer, was employed for the mass spectrometric identification of QAP. This facilitated the exploration of critical information, including the amino acid sequences and molecular weights of the associated proteins. Furthermore, a series of experiments were conducted to investigate the mode of action of QAP on C. albicans, and the molecular mechanism underlying QAP’s inhibitory effect on C. albicans was elucidated through the integration of RNA sequencing technology.

Materials & Methods

Plant materials

The quinoa seeds (Cheng Li No.1) utilized in this study were supplied by the Key Laboratory of Coarse Cereal Processing, Ministry of Agriculture and Rural Affairs, Chengdu University (Fig. 1). This quinoa cultivar, meticulously developed by Chengdu University, is characterized by a high protein content. Specifically, the protein content of this variety is approximately 147.38 g/kg, while the starch content is approximately 563.36 mg/g.

Figure 1 Phenotypic characteristics of Cheng Li NO.1 seeds.

(A) Color. (B) Shape (scale bar = 1.00 mm).

Strains and culture conditions

C. albicans (SIIA 2284) was provided by the Department of Urology at the Affiliated Hospital of Chengdu University. The strain was initially cultured in Sabouraud Dextrose Broth (SDB) and subsequently activated by incubation for 18 h at 180 rpm and 37 °C.

Preparation of QAP

Quinoa seeds were initially screened to eliminate impurities, followed by drying and grinding into a fine powder. The resultant powder was then passed through a 100-mesh sieve and subsequently immersed in petroleum ether for defatting. Post-defatting, the quinoa flour was subjected to extraction at a 1:2 ratio using a 7 mM phosphate extraction buffer (pH 7.0, containing 10 mM NaCl and 1 mM EDTA) at 4 °C for 2 h. The extraction mixture was centrifuged at 4 °C and 10,000 rpm for 20 min, and the supernatant was collected as the crude QAP extract. The protein content of the crude QAP extract was quantified using the protein quantification assay kit according to the instructions provided by the Nanjing Jiancheng Bioengineering Institute. Solid ammonium sulfate was finely ground, and the crude extract of QAP was subjected to stepwise salting out using a 10% gradient. Following incubation at 4 °C for 40 min, the precipitates obtained from each fraction were dissolved in a minimal volume of phosphate extraction buffer (pH 7.0) containing 20 mmol/L. These solutions were subsequently dialyzed against a phosphate-buffered saline (PBS) buffer solution (20 mmol/L). Finally, the QAP solution was concentrated to a uniform volume, and the protein concentration was measured and adjusted as necessary. The inhibitory effects of QAP on C. albicans were evaluated using the Oxford cup-hole method. Under aseptic conditions, Oxford cups were placed into Petri dishes. Once the Sabouraud dextrose agar medium had cooled to approximately 50 °C, a suspension of C. albicans at a concentration of 3 × 108 CFU/mL was added at a volume ratio of 1,000:1 and thoroughly mixed before the medium was poured into the Petri dishes. Following the solidification of the medium, the Oxford cups were meticulously removed using sterilized tweezers. Subsequently, 200 µL of QAP extracts, which were prepared under different ammonium sulfate saturation conditions, were added into the cavities left by the Oxford cups. The Petri dishes were then incubated at 37 °C for 24 h. Physiological saline served as the control group, while the clinical drug fluconazole (FLC) functioned as the positive control group. The criterion for assessing the antifungal efficacy was predicated on the external diameter of the Oxford cup. Specifically, the diameter of the inhibition zone was quantified using the cross-streaking technique, with the center of the indentation created by the Oxford cup serving as the reference point for measurement. A diameter exceeding 7.8 mm was considered indicative of a significant antifungal effect. Conversely, diameters below this threshold were deemed to demonstrate no antifungal activity, as the growth and reproduction of C. albicans were not effectively inhibited. In accordance with clinical pharmacological guidelines, the concentration of FLC was maintained at 160 µg/mL.

Quinoa bioactive proteins are separated and purified using gel chromatography

The preparation of the QAP involved utilizing Sephadex G-75 and Sephadex G-50 as packing materials within a chromatographic column with dimensions of 2.0 cm × 40 cm. An appropriate quantity of Sephadex G-75 powder was accurately weighed and transferred into a 500 mL beaker. Subsequently, deionized water or elution buffer, in a volume 7–10 times that of the Sephadex G-75, was added. The resulting (wet) gel was subjected to a digital constant temperature water bath set near boiling for 30 min. After this thermal treatment and cooling, the upper layer containing gel debris and impurities was carefully removed. Additional deionized water or elution buffer was added, and the mixture was equilibrated at ambient temperature for a duration of 2 h to ensure complete swelling. Subsequently, the supernatant was decanted from the swollen Sephadex G-75 gel, and a volume of phosphate extraction solution equivalent to 1–2 times the gel volume was added to form a suspension. This suspension was then gently agitated and transferred into the column using a glass rod to facilitate slurry packing. Initially, a small portion of the gel was permitted to settle within the column prior to opening the outlet valve. The slurry packing process continued was carried out by incrementally adding and uniformly distributing the gel until a settling height of approximately 35 cm was attained, at which point the outlet valve was closed. Following the formation of the gel column, the extraction solution was added into the elution bottle. The gel column was equilibrated by passing three column volumes of the extraction solution through the column at a flow rate of 1.3 mL/min. The procedure for handling Sephadex G-50 filler was identical to that employed for Sephadex G-75.

Protein mass spectrometry

The supernatants containing quinoa proteins, precipitated at varying levels of ammonium sulfate saturation, were analyzed using SDS-PAGE electrophoresis followed by silver staining. For protein identification, the QAP extracted under different ammonium sulfate saturation conditions, along with those purified via Sephadex G-75 and Sephadex G-50 gel filtration chromatography, were subjected to LC-MS/MS analysis (Triple ToF 5600+AB-SCIEX). The resulting peptide sequences were identified using ProteinPilot software. Subsequently, the proteins were classified via the InterProScan (IPR)-based domain annotation method.

Quantification of QAP’s minimum inhibitory concentration (MIC)

The QAP solution (with concentrations ranging from 728 µg/mL to 2.8 µg/mL) and the FLC solution (with concentrations from 256 µg/mL to 0.5 µg/mL) were prepared by serial two-fold dilution. According to the method of Xie et al. (2024), the antifungal activity of the QAP solution was determined using the Oxford cup punching method, consistent with the procedure detailed in the “Preparation of QAP” section. Using a sterile pipette, 200 µL of each concentration of the QAP and FLC solutions was aspirated and then added into the holes created by the Oxford cups to establish test wells. An equivalent volume of sterile physiological saline served as a blank control. The culture dishes were then sealed and incubated at a constant temperature of 37 °C for 24 h. Following incubation, observations were made regarding the areas surrounding the test wells in the culture dishes. The minimum inhibitory concentration (MIC) for the QAP solution was identified as the lowest concentration that resulted in an inhibition zone exceeding 7.8 mm in diameter. Similarly, the MIC for the FLC solution was determined as the lowest concentration that produced an inhibition zone with a diameter greater than 7.8 mm.

Growth kinetics of C. albicans under QAP

Under sterile conditions, 10 µL of C. albicans solution (SIIA 2284, 3 × 108 CFU/mL) was extracted from the blank control group and transferred to a sterilized 96-well plate. Subsequently, 190 µL of RPMI-1640 complete medium was added and thoroughly mixed. For the QAP treatment group, 10 µL of C. albicans solution was obtained from a sterilized 96-well plate and combined with 100 µL of QAP solution (0.78 mg/mL) and 90 µL of RPMI-1640 complete medium. The mixture was meticulously blended before being incubated at 37 °C with agitation at 80 rpm for growth monitoring. The growth curve was constructed by measuring A600 every two hours using a Multifunctional Microplate Reader (Synergy HTX; BioTek, Winooski, VT, USA).

Determination of alkaline phosphatase (AKP) activity

To explore the inhibitory mechanism of QAP against C. albicans, we assessed the activity of alkaline phosphatase (AKP), a commonly utilized indicator for evaluating the cell wall damage or integrity (Song et al., 2018). The activated C. albicans suspension was centrifuged at 4,000 rpm for 2 min and the supernatant was removed. The resultant pellet was subsequently re-suspended in sterile phosphate-buffered saline (PBS) to attain a concentration of 3 × 108 CFU/mL. Subsequently, 200 µL of the C. albicans suspension was added to a sterile conical flask containing 20 mL of SDB solution, followed by the addition of 20 mL of QAP for treatment. In the blank control group, an equal volume of SDB solution was added. The cultures were incubated at 37 °C and 180 rpm for 8 h. Post-incubation, the supernatant was collected by centrifugation at 10,000 rpm for 10 min. The extracellular AKP activity was then determined according to the instructions provided with the AKP assay kit (Nanjing Jiancheng Bioengineering Institute, Nanjing, China).

Activity analysis of Succinate dehydrogenase (SDH), Ca2+-Mg2+- ATPase and Catalase (CAT)

The conditions for sample processing were maintained in accordance with AKP activity determination, whereas the control group was replaced with PBS. The activity of SDH, CAT and Ca2+-Mg2+-ATPase was determined by an Enzyme-linked Immunoassay Kit (Jiangsu Meimian industrial Co., Ltd.) according to the manufacturer’s protocol. The specific techniques employed for measurement are provided in Text S1.

C. albicans hyphal morphology assay

The nutrient-rich Spider medium, which utilizes mannitol as its carbon source, is commonly employed to facilitate morphogenesis (Véronique et al., 2024). For the current study, the Spider agar medium was utilized to evaluate the impact of QAP on the yeast-to-hyphal transition in C. albicans (Chu et al., 2025; Sun et al., 2023). C. albicans was cultured on the Spider agar medium with varying treatments: a negative control (0 mg/mL), a test group with 0.78 mg/mL QAP, and a positive control group with 160 µg/mL FLC. The cultures were incubated at 37 °C for 96 h. Following incubation, the morphological characteristics of the C. albicans colonies were documented.

Observation of cell morphology by scanning electron microscope (SEM)

Under aseptic conditions, activated C. albicans cells were centrifuged, and the C. albicans suspension was adjusted to a concentration of 3 × 108 CFU/mL. Two sterilized 150 mL Erlenmeyer flasks were utilized: Flask 1 contained 40 mL of the C. albicans suspension and 20 mL of the QAP component solution at a concentration of 0.78 mg/mL; Flask 2 contained 40 mL of the C. albicans suspension and 20 mL of phosphate extraction liquid.

Following an 8-h incubation period at 37 °C with agitation at 180 rpm, the culture was aseptically transferred to sterilized centrifuge tubes and subjected to centrifugation at 4,000 rpm for 1 min. The resulting supernatant was carefully removed, and the fungal organisms were harvested. Subsequently, the fungal cells underwent three washes with pre-sterilized PBS to eliminate any residual contaminants. The washed cells were subsequently fixed in a 2.5% glutaraldehyde solution at 4 °C under light-excluded conditions for 2 h to ensure optimal preservation of cellular structures. Post-fixation, the samples were subjected to a graded ethanol dehydration series (30%, 70%, and 100%) for approximately 10 min each at room temperature. Following dehydration, the samples were air-dried overnight on sample holders secured with double-sided adhesive tape. Subsequently, gold sputter coating was performed utilizing an HVS-GB vacuum evaporator under meticulously controlled conditions. Following this, the samples were subjected to localization and imaging using a Sirion 200 scanning electron microscope.

Isolation of total RNA and preparation of cDNA libraries

Six 150 mL Erlenmeyer flasks with uniform specification, were prepared and labeled sequentially from 1 to 6. Each flask was filled with 50 mL of SDB and subjected to sterilization at 121 °C for 20 min, followed by cooling for subsequent experimental procedures. Under strictly aseptic conditions, a single colony of C. albicans was inoculated into each flask. The inoculated cultures were then incubated at 37 °C with agitation set at 180 rpm for 18 h. Following the initial incubation phase, Flask 1 was supplemented with 50 mL of QAP solution at a concentration of 0.45 mg/mL; Flasks 2 and 3 underwent the same treatment as Flask 1. Concurrently, Flask 4 received an addition of 50 mL of phosphate extraction liquid, with Flasks 5 and 6 being treated similarly to Flask 4. Subsequently, all six flasks were reincubated under consistent conditions (37 °C, 180 rpm) for an additional 8-h duration. Upon the conclusion of this secondary incubation phase, the contents of each flask were meticulously transferred into sterile centrifuge tubes and centrifuged at 4,000 rpm for 2 min to pellet the biomass. The resultant cell pellets were flash-frozen in liquid nitrogen and were promptly stored at −80 °C for subsequent analysis. The RNA was extracted from C. albicans cells following a standardized protocol and subjected to stringent quality control assessment of RNA integrity using the Agilent 2100 Bioanalyzer. The fragmented mRNA was used as a template for the synthesis first-strand cDNA, employing random oligonucleotides in the M-MuLV reverse transcriptase system. Subsequently, RNase H was utilized to degrade the RNA strand, facilitating the synthesis of second-strand cDNA using dNTPs in the DNA polymerase I system. Following terminal repair, adenylation, and the ligation of sequencing adaptors, the purified double-stranded cDNA was subjected to size selection (approximately 250–300 base pairs) using AMPure XP beads. The selected cDNA fragments were then amplified via polymerase chain reaction (PCR) to construct the library. The statistical power of this experimental design, as calculated using RNASeqPower, is 0.83.

Data analysis following illumina sequencing

Following a comprehensive library inspection, various libraries are categorized according to the prerequisites for optimal concentration and target data volume necessary for Illumina sequencing technology. Amplification is facilitated by the introduction of four fluorescently labeled deoxynucleotide triphosphates (dNTPs), DNA polymerase, and adapter primers into the flow cell. Each incorporation of a fluorescently labeled dNTP generates a corresponding fluorescence signal. This signal is subsequently captured and processed by computer software to produce sequencing peaks, thereby yielding the sequence information of the desired fragments. The acquired sequence data is subsequently subjected to a quality assessment protocol, wherein statistical analyses are conducted to identify discrepancies between the raw data and the quality-controlled processed data.

Following the completion of quality control, the high-quality sequences were aligned to the reference genome using HISAT2 software (Mortazavi et al., 2008). Subsequently, the mapped sequences were assembled to the genome utilizing StringTie software (Pertea et al., 2015). A comparative analysis with existing gene annotations was then conducted using GffCompare software to identify unannotated transcript regions and discover novel transcripts or genes within this species. To ensure accurate assessment of gene expression levels (Garber et al., 2011), FPKM values for each gene were calculated using StringTie (Bray et al., 2016). Subsequently, the Euclidean distance metric was employed to assess the correlations among samples for the purpose of hierarchical clustering. Following the clustering process, a heatmap was constructed, wherein the intensity of color denotes the variance in gene expression patterns between samples, with lighter shades indicating greater differences and darker shades representing smaller differences. The resulting dendrogram illustrates the degree of similarity among samples, with proximal branches signifying higher similarity, and samples exhibiting greater similarity tending to cluster more closely together.

Ultimately, the DESeq2 software was utilized to conduct a significance analysis of variations in gene expression. Typically, a gene demonstrating a fold change (FC) greater than two between two sample groups is considered to exhibit significant differential expression.

Quantitative real-time PCR

The expression levels of regulatory genes in C. albicans inhibited by QAP were quantitatively assessed using real-time fluorescence quantitative polymerase chain reaction (qRT-PCR) technology. This procedure encompassed RNA extraction and purification, followed by reverse transcription into complementary DNA (cDNA) utilizing the FastKing RT Kit (With gDNase) FastKing cDNA, in accordance with the manufacturer’s protocol provided by TianGen Biotech (Beijing) Co., Ltd. The specific genes analyzed in this study, along with their corresponding primers, are enumerated in Table S1. Quantitative real-time PCR (qRT-PCR) assays were conducted in triplicate utilizing a qTOWER  ˆ 3G Real-Time PCR system (Analytik Jena AG, Jena, Germany). The thermal cycling protocol comprised an initial denaturation step at 95 °C for 30 s, followed by 40 cycles of denaturation at 95 °C for 5 s and annealing/extension at 60 °C for 30 s. The actin gene (ACT1) served as the reference gene, and relative gene expression levels were quantified using the 2 ˆ−ΔΔCT method.

Statistical analysis

To elucidate the functionality of specific genes, an extensive search was conducted within the Gene Ontology (GO) and Kyoto Encyclopedia of Genes and Genomes (KEGG) databases (Table S2). Subsequently, a comprehensive analysis of GO functional enrichment (Young et al., 2010) and KEGG pathway enrichment was performed on the set of DEGs using the clusterProfiler software (Table S3). Following the Z-score normalization of the expression levels of the identified DEGs, clustering analysis was undertaken to group genes exhibiting similar expression patterns. Ultimately, a clustering heatmap was generated using the Wekemo Bioincloud Platform (http://www.bioincloud.tech).

The data collected in this study were systematically organized using Excel 2016 software and subsequently visualized through GraphPad Prism 9 (version 9.5.0) and Cytoscape (version 3.9.1). Statistical comparisons between two independent samples were analyzed using the t-test, while differences among three or more independent samples were assessed via one-way ANOVA, supplemented by Duncan’s multiple range test. The significance threshold was established at P < 0.05. These statistical analyses were performed using the pertinent functions available in SPSS 27 software. The experiment was conducted in triplicate, and the results are presented as x ± n. Asterisks indicate statistically differences between control and treatment groups. The RNA-seq data have been deposited in the NCBI Sequence Read Archive (SRA) database (accession number: PRJNA1166727).

Results

Isolation and purification of antifungal QAP

To elucidate the final saturation range of ammonium sulfate necessary for QAP precipitation to inhibit C. albicans, an investigation was conducted within the 0–30% ammonium sulfate saturation range. Although QAP demonstrated inhibitory effects against C. albicans within this range, there were no statistically significant differences in the diameters of the inhibition zones (p > 0.05). This suggests that the proteins precipitated under these conditions were likely contaminants (Fig. 2). As the ammonium sulfate saturation increased to the range of 30%–70%, the QAP precipitated within this range exhibited significant variations in inhibitory activity against C. albicans (Fig. 3). However, at a saturation level of 80%, the inhibitory effect markedly weakened, suggesting that the majority of QAP had been fully precipitated under these conditions. Proteomic analysis of protein precipitation at each saturation showed that there were 644, 716, 690, 647, 643 and 622 proteins in different ammonium sulfate saturation segments of 20%–30%, 30%–40%, 40%–50%, 50%–60%, 60%–70% and 70%–80%, respectively. The results showed that most of the active proteins of quinoa could be effectively separated with different saturation of ammonium sulfate, and there were about 149 common proteins in the 30%–80% ammonium sulfate salting phase (Fig. 4).

Figure 2 The impact of ammonium sulfate saturation on QAP’s (quinoa active protein) inhibition of Candida albicans (C. albicans).

Different lowercase letters indicate significant differences between different treatment. Error bars indicate means ± SD (n = 3). Different lowercase letters on bars indicate significant differences determined at the level of p < 0.05 by Duncan’s multiple range test.

Figure 3 (A–B) Inhibition of C. albicans by ammonium sulfate precipitation at various saturation levels.

Scale bar = 10 mm.

Figure 4 Upset diagram of proteins in different ammonium sulfate saturation segments.

S30, 30% ammonium sulfate saturation for QAP isolation. S40, 40% ammonium sulfate saturation for QAP isolation. S50, 50% ammonium sulfate saturation for QAP isolation. S60, 60% ammonium sulfate saturation for QAP isolation. S70, 70% ammonium sulfate saturation for QAP isolation. S80, 80% ammonium sulfate saturation for QAP isolation. About 149 common proteins were found in the 30%–80% ammonium sulfate salting phase.

Analyses of QAP by SDS-PAGE at different saturation of ammonium sulfate

The supernatants containing QAP, precipitated at varying ammonium sulfate saturation levels, were analyzed using SDS-PAGE electrophoresis. As illustrated in Fig. 5, the molecular weights of quinoa proteins were observed to range from approximately 14.4 to 116.0 kDa. Notably, as the ammonium sulfate saturation increased, the proteins within the 18.4–25 kDa range in the supernatant showed a significant decrease. Meanwhile, the antifungal activity of the precipitated proteins exhibited a progressive enhancement. This suggests that the antifungal proteins are likely within this molecular weight range, closely aligning with the molecular mass (24 kDa) of the antimicrobial protein discovered by Wang et al. (2011) from the seed of Cynanchum komarovii Al Iljinski. When considered alongside the separation outcomes obtained from Sephadex G-75 chromatography, the relative abundance of proteins within the specified range was observed to increase. This observation further substantiates the hypothesis that the antimicrobial proteins are situated within the 18.4–25 kDa range, corroborating the results of previous proteomic analyses (Vilcacundo, Martínez-Villaluenga & Hernández-Ledesma, 2017). Furthermore, research conducted by Osman, Mahgoub & Sitohy (2013) demonstrated that the subunits of soybean 11S globulin exhibit antimicrobial activity. These findings indicate that ammonium sulfate fractionation serves as an effective technique for isolating QAP with inhibitory properties against C. albicans, while simultaneously distinguishing these compounds from other proteins.

Figure 5 SDS-PAGE electrophoresis analysis QAP.

lane1, Saturated range of 20–30% ammonium sulphate; lane2, saturated range of 30–40% ammonium sulphate; lane3, saturated range of 40–50% ammonium sulphate; lane4, saturated range of 50–60% ammonium sulphate; lane5, saturated range of 60–70% ammonium sulphate; lane6, saturated range of 70–80% ammonium sulphate; lane7, Sephadex G-75 purification of proteins; lane8, crude extract of QAP obtained from quinoa seeds using phosphate extraction buffer.

Purification of QAP by dextran gel chromatography

QAP solutions, purified using Sephadex G-75 and Sephadex G-50 columns, were evaluated for their antifungal activity against C. albicans on Sabouraud dextrose agar medium. The QAP solution derived from Sephadex G-75 exhibited a concentration of 0.78 mg/mL, whereas the solution from Sephadex G-50 had a concentration of 0.4 mg/mL. In comparison to the results obtained from purification using a Sephadex G-75 column, the spectrum of active protein fractions in quinoa was further refined when purified with a Sephadex G-50 column (Fig. 6).

Figure 6 QAP protein classification via the InterProScan (IPR)-based domain annotation method.

(A) Sephadex G-75. (B) Sephadex G-50.

Proteomic analysis revealed that 18 proteins shared between the fractions obtained through Sephadex G-75 and Sephadex G-50 purification (Fig. 7). It was observed that, despite a significant decrease in the concentration of QAP, an increase in the diameter of the inhibition zones was observed between the two groups (p < 0.05). Moreover, when compared to the clinical drug FLC, the inhibitory zone diameter of QAP, purified by Sephadex G-50, was larger than that of FLC (Figs. 8A and 8B).

Figure 7 Venn diagram for Sephadex G-75 and Sephadex G-50 gel chromatography proteins.

Figure 8 Antifungal efficacy of QAP purified by Sephadex G-75/Sephadex G-50.

Different lowercase letters indicate significant differences between different treatment samples (p < 0.05). (A) Oxford cup method for evaluating QAP inhibition of C. albicans. (a) Purification of QAP by Sephadex G-50, (b) purification of QAP by Sephadex G-75, (c) positive control group: FLC, (d) blank control group: physiological saline. (B) Column chart of inhibition zone diameter of QAP-treated C. albicans. Error bars indicate means ± SD (n = 3). Different letters on bars indicate significant differences determined at the level of p < 0.05 by Turkey’s test. Scale bar = 10 mm.

Determination of MIC of QAP

The MIC of QAP against C. albicans was determined to be 182 µg/mL (Table 1), with an associated antifungal zone diameter of 1.10 ± 0.04 cm. Meanwhile, for the positive control group, the MIC was found to be 16 µg/mL, with a corresponding antifungal zone diameter of 1.84 ± 0.16 cm (Table 2). The effect diagrams of antifungal activity are presented in Data S1.

Table 1 Minimum inhibitory concentration (μg/mL) of quinoa active protein (QAP).

Concentration (μg/mL)	728	364	182	91	45.5	22.7	11.3	5.6	2.8	
Inhibition zone diameter (mm)	3.55 ± 0.13a	2.96 ± 0.10b	1.10 ± 0.04c	–	–	–	–	–	–	
Notes.

Data are presented as mean ± SD.

Different lowercase letters among treatment groups indicate that there are significant differences in the antifungal effects of QAP at different concentrations (p < 0.05). n = 3.

“–” indicates the absence or indistinct presence of an inhibition zone.

Table 2 Minimum inhibitory concentration (μg/mL) of the FLC.

Concentration (μg/mL)	256	128	64	32	16	8	4	2	1	0.5	0.25	
Microbial inhibition effect	3.81 ± 0.04a	3.16 ± 0.19b	2.81 ± 0.09c	2.46 ± 0.06d	1.84 ± 0.16e	–	–	–	–	–	–	
Notes.

Data are presented as mean ± SD.

Different lowercase letters among treatment groups indicate that there are significant differences in the antifungal effects of FLC at different concentrations (p < 0.05). n = 3.

“–” indicates the absence or indistinct presence of an inhibition zone.

QAP caused cell wall damage in C. albicans

As shown in Fig. 9, the extracellular AKP activity of C. albicans treated with 0.78 mg/mL of QAP was 0.98 ± 0.09 U/mL (p < 0.05), which is higher than that of the control group (0.62 ± 0.09 U/mL). Therefore, we concluded that QAP has the potential to induce cell wall damage in C. albicans.

Figure 9 Changes of extracellular alkaline phosphatase activity of C. albicans after QAP treatment.

CK, blank control group; QAP, QAP treatment group (0.78 mg/mL). Error bars indicate means ± SD (n = 3). *P < 0.05.

QAP exhibits inhibitory effects on the growth and reproduction of C. albicans

As illustrated in Fig. 10, the initial five-hour period was characterized by relatively sluggish growth of C. albicans, with no significant difference in optical density (OD) observed between the two groups. Between 5 and 20 h, the OD of the control group increased rapidly, whereas the growth rate of the OD in the QAP group was markedly lower compared to the control group during this interval. By 30 h, both groups reached a stable state, with the OD of the control group recorded at 1.400 ± 0.008, while the OD of the QAP group was 0.621 ± 0.010. Throughout the duration of the study, the OD of the QAP group consistently remained lower than that of the control group.

Figure 10 Growth dynamics of C. albicans in the presence of QAP.

CK, blank control; QAP, QAP treatment group (0.78 mg/mL). Error bars indicate means ± SD (n = 3).

Activity analysis of succinate dehydrogenase (SDH), Ca2+-Mg2+- ATPase and catalase (CAT)

When C. albicans was treated with QAP at concentrations of 0.78 mg/mL, 0.52 mg/mL, and 0.26 mg/mL for 8 h, the succinate dehydrogenase (SDH) activities were recorded as 334.4 ± 5.8 U/L, 341.8 ± 8.8 U/L, and 363.0 ± 6.5 U/L, respectively, while the corresponding calcium-magnesium ATPase (Ca2+-Mg2+-ATPase) activities were 92.84 ± 0.97 IU/L, 92.85 ± 2.08 IU/L, and 87.10 ± 1.20 IU/L. Concurrently, in the positive control group treated with fluconazole (FLC) at concentrations of 320 µg/mL, 160 µg/mL, and 80 µg/mL, the SDH activities were measured at 317.5 ± 8.9 U/L, 319.3 ± 5.0 U/L, and 345.0 ± 5.9 U/L, with Ca2+-Mg2+-ATPase activities of 75.21 ± 0.30 IU/L, 79.99 ± 0.61 IU/L, and 77.69 ± 1.31 IU/L. Compared to the blank control group, which exhibited an SDH activity of 407.8 ± 8.1 U/L and a Ca2+-Mg2+-ATPase activity of 98.56 ± 1.59 IU/L, both the QAP-treated and FLC-treated groups demonstrated statistically significant reductions in SDH and Ca2+-Mg2+-ATPase activities (P < 0.0001). Through comprehensive data analysis, it is evident that QAP affects the Ca2+-Mg2+-ATPase located on the cell membrane and the intracellular succinate dehydrogenase (SDH) of C. albicans, thereby disrupting intracellular homeostasis and energy metabolism. This disruption is hypothesized to result from modifications in enzymatic activities, which are crucial for maintaining the normal physiological state of the cell (Figs. 11A and 11B).

Figure 11 The impact of different concentrations of QAP and FLC on enzyme activities in C. albicans.

(A–C) Enzyme-linked immunosorbent assay (ELISA) for the detection of succinate dehydrogenase (SDH), Ca2+-Mg2+-ATPase and catalase (CAT) activity in C. albicans. Cells were treated with QAP (QAP1: 0.78 mg/mL, QAP2: 0.52 mg/mL, QAP3: 0.26 mg/mL), and PBS was used as the negative control, and FLC was used as the positive control (FLC1: 0.32 mg/mL, FLC2: 0.16 mg/mL, FLC3: 0.08 mg/mL). Data represent mean ± SD, n = 4. ****P < 0.001.

Moreover, an increase in the CAT activity of C. albicans was observed following treatment with varying concentrations of QAP and FLC. The CAT activity of the negative control group was measured at 14.03 ± 0.36 U/mL, whereas the CAT activities of the QAP-treated group at concentrations of 0.78 mg/mL, 0.52 mg/mL, and 0.26 mg/mL were 19.92 ± 0.44 U/mL, 17.42 ± 0.37 U/mL, and 18.79 ± 0.52 U/mL, respectively. Concurrently, the CAT activities of the FLC-treated group at concentrations of 320 µg/mL, 160 µg/mL, and 80 µg/mL were 17.34 ± 0.18 U/mL, 16.32 ± 0.22 U/mL, and 18.25 ± 0.41 U/mL, respectively, with these differences being statistically significant. This experimental data suggests that QAP has disrupted the normal metabolic activities of C. albicans, potentially affecting energy metabolism and membrane functions, as evidenced by alterations in SDH and Ca2+-Mg2+-ATPase activities. Consequently, this disruption results in increased intracellular oxidative stress, leading C. albicans to enhance its CAT activity as a compensatory response to mitigate the excessive reactive oxygen species generated due to the disrupted metabolic activities (Fig. 11C).

The effects of QAP on yeast-to-hyphal transition

The ability of C. albicans to transition from the yeast form to the hyphal form is considered one of its most crucial virulence factors. As depicted in Fig. 12A, the negative control group formed dense long hyphae when cultured on the Spider agar medium. However, upon treatment with either QAP (0.78 mg/mL) or FLC (160 µg/mL), the colonies formed by C. albicans cells displayed irregular edges, and produced only a limited number of hyphae (Figs. 12B and 12C). This could be attributed to the diffusion of QAP or FLC in the Spider agar medium (Yang et al., 2024) potentially disrupting the local concentration gradients of nutrients or signaling molecules essential for the normal hyphal growth of C. albicans, leading to the observed irregular colony edges and reduced hyphal production. These findings suggest that QAP effectively inhibits the transition of C. albicans from the yeast form to the hyphal form in Spider agar medium.

Figure 12 QAP inhibits the hyphal growth of C. albicans.

(A) Negative control group, (B) QAP treatment group (0.78 mg/mL), (C) positive control group (160 µg/mL). Red arrows pointed to the hyphae. Scale bar = 10 mm.

QAP affects C. albicans colony morphology

This study employed SEM to investigate the effects of QAP on the morphology of C. albicans. The results indicated that the C. albicans displayed plump and rough cell bodies undergoing division and growth in the control group. While C. albicans displayed smoother cell surfaces characterized by distinct cracks and a marked decrease in budding sites in the treated group with 0.78 mg/mL of QAP (Fig. 13).

Figure 13 Scanning electron microscope study of C. albicans morphological structure after QAP treatment.

(A, B) Untreated C. albicans exhibits a robust morphology, characterized by numerous and prominent budding sites. (C, D) C. albicans exposed to QAP at a concentration of 0.78 mg/mL for 8 h exhibited cellular contraction, characterized by the presence of prominent fissures on certain cells, and a progressive flattening of budding sites (red arrows).

Results of illumina sequencing and assembly

Following the construction and quality assessment of the cDNA library, various library combinations were subjected to Illumina sequencing according to the desired effective concentration and target data volume. Subsequently, the raw sequencing data for each sample were then subjected to quality control using the fastp software. Ultimately, 38.09 G clean reads were obtained, with a GC content of 36.77% and a Q30 value of 93.78% (Table 3). These metrics satisfy the requirements for transcriptome analysis.

Table 3 Sequencing raw data and filtered data quality statistics.

Sample	Raw reads	Seq type	Raw base (G)	Clean reads	Q20 (%)	Q30 (%)	GC content (%)	
CK1	20,142,685	(150:150)	6.04	17,941,475	97.91	93.86	36.98	
CK2	20,833,730	(150:150)	6.25	19,016,542	97.72	93.5	36.95	
CK3	22,883,586	(150:150)	6.87	21,088,193	97.96	93.97	36.86	
Treat1	20,799,991	(150:150)	6.24	19,398,514	97.97	94	36.74	
Treat2	22,133,645	(150:150)	6.64	20,217,028	97.66	93.36	36.64	
Treat3	20,157,631	(150:150)	6.05	18,761,634	98	94.01	36.47	

In this study, the correlation among samples was evaluated utilizing Euclidean distance, and a hierarchical clustering heatmap was generated to analyze the gene expression level correlations across various samples. The clustering of three biological replicates within each experimental and control group revealed significant differences in expression profiles between the groups, while also demonstrating a high degree of correlation among replicates within each group (Fig. 14A). A comprehensive analysis revealed that 6,263 genes were expressed across all samples. Differential expression analysis was performed using the DESeq2 software package, employing stringent screening criteria of |log2(fold change)| > 1 and an adjusted p-value (p. adj) < 0.05. As a result, 1,413 genes exhibited significant differential expression, with 892 genes up-regulated and 521 genes down-regulated (Figs. 14B and 14C).

Figure 14 Heat maps for correlation, differential gene volcano maps, and enrichment analysis maps.

(A) Euclidean distance clustering heatmap: this heatmap quantifies the inter-sample correlation, demonstrating that samples with higher similarity tend to cluster together. (B, C) Analysis of differential gene expression between untreated C. albicans and QAP-treated C. albicans, highlighting the number of upregulated and downregulated genes. (D, E) Gene Ontology (GO) molecular function and Kyoto Encyclopedia of Genes and Genomes (KEGG) enrichment analysis. (F) Heatmap illustrating differential gene expression derived from selected genes.

Unigene function annotation

A comprehensive annotated and classified analysis of differentially expressed genes (DEGs) was performed using Gene Ontology (GO) and Kyoto Encyclopedia of Genes and Genomes (KEGG) analyses to uncover the underlying biological mechanisms and pathways.

GO analysis revealed that, compared to the control group, the QAP-treated group exhibited significant enrichment in GO terms associated with the cell wall, cellular respiration, reactive oxygen species metabolism, ion transport, ergosterol biosynthesis, and transmembrane transport proteins (Fig. 14D). To further elucidate the molecular mechanisms underlying the inhibitory effects of QAP on C. albicans, these DEGs were further subjected to KEGG pathway annotation and enrichment analysis. The results revealed significant enrichment in pathways including DNA replication proteins, oxidative phosphorylation, the tricarboxylic acid (TCA) cycle, and glycosyltransferase (Fig. 14E). Subsequently, a heatmap was generated to illustrate differential gene expression based on the DEGs within the selected pathways (Fig. 14F).

Through transcriptome analysis, we identified DEGs within specific pathways for subsequent investigation of protein–protein interaction (PPI) networks using the STRING online platform (https://string-db.org/). The selected DEGs and their corresponding analysis results are listed in Table S4. Subsequently, Cytoscape software was utilized to perform topological analyses on the resulting PPI network graph, with node size, coloration, and intensity indicating their respective degree values. Our findings revealed that SDH2 (Succinate dehydrogenase 2) and ACO1 (Aconitase 1) are pivotal proteins (Fig. 15). These proteins jointly participate in the TCA cycle and play a crucial role in the energy metabolism of C. albicans. Additionally, qRT-PCR validation of the expression levels of 13 selected differentially expressed genes revealed consistent trends with our initial transcriptome analysis, and all the results exhibited statistically significant differences (P < 0.05). Specifically, the expression levels of genes ZRT1, SIM1, ALS4, and PRA1 showed a significant increase (P < 0.05) following the treatment with quinoa active proteins. Conversely, the expression levels of the remaining 11 genes showed a significant decrease (P < 0.05). Notably, the expression levels of FBA1, CST20, ERG1, and SDH2 showed an extremely significant decrease after being treated with QAP (P < 0.0001), with the expression level of SDH2 decreasing by more than threefold (Fig. 16).

Figure 15 PPI network.

The genes identified through the previously outlined procedures, which are associated with hyphal development, cell wall integrity, cell membrane function, cellular respiration, and energy metabolism, were utilized to construct a protein–protein interaction (PPI) network using the STRING database. In this network, genes are represented as nodes, while their interactions are depicted as edges.

Figure 16 (A–M) The expression profile of C. albicans-associated genes following treatment with QAP.

Data represent mean ± SD, n = 3. *P < 0.05, **P < 0.01, ***P < 0.001, ****P < 0.001.

Discussion

Quinoa seeds are abundant in proteins, polyphenols, and various bioactive compounds. In contemporary nutrition science, plant-derived proteins are increasingly recognized for their significant contributions to human health. Proteins sourced from plant seeds have attracted considerable interest due to their antioxidative, antihypertensive, and lipid-lowering properties. Additionally, research has indicated that proteins extracted from diverse plant seeds possess antimicrobial activities. For instance, antifungal proteins derived from Job’s tears seeds and antimicrobial peptides obtained from soybean proteins have been shown to inhibit the growth of Aspergillus niger. However, there are no published reports indicating that quinoa protein inhibits C. albicans.

Multidrug-resistant infections caused by C. albicans, along with the significant clinical side effects associated with such infections, present pervasive challenges that substantially impact global mortality and morbidity rates, while also imposing considerable economic burdens (Islam et al., 2024). In this study, active proteins with inhibitory effects against C. albicans were isolated and purified from quinoa using traditional ammonium sulfate precipitation. The inhibitory activity of the purified QAP was evaluated using the Oxford cup method, revealing a MIC of 182 µg/mL against C. albicans. Fractionation with ammonium sulfate gradients, ranging from 30% to 70% saturation, was effective in extracting QAP. Proteomic analysis identified a total of 149 quinoa proteins extracted at different levels of ammonium sulfate saturation, out of an estimated 40,000 proteins present in quinoa seeds. This method thus provides a preliminary strategy for the isolation and purification of QAP.

To further purify the active proteins derived from quinoa, gel filtration chromatography Sephadex G-75 and Sephadex G-50 was subsequently implemented. According to the manufacturer’s specifications, Sephadex G-75 and Sephadex G-50 are capable of separating proteins within the molecular weight ranges of 3–80 kDa and 1.5–30 kDa, respectively. Despite significant reduction in the quantity of purified QAP and a gradual decrease in protein concentration, there was an observed increase in the diameter of the inhibitory zone against C. albicans. Integrating data from proteomic analysis, it can be preliminarily inferred that among the 18 proteins common to both fractions, the QAP exhibiting inhibitory effects against C. albicans were identified as 11S globulins.

By quantifying the extracellular AKP activity of C. albicans, the findings indicated a significant increase in extracellular AKP levels following treatment with QAP. AKP is localized between the cell wall and the membrane of C. albicans. An increase in cell wall permeability, or damage to the cell wall, results in the leakage of AKP into the extracellular environment. This observation suggests that QAP impacts the cell wall integrity of C. albicans.

To further elucidate the mechanism by which QAP inhibits C. albicans, we investigated the effects of different concentrations of QAP and FLC on the activities of SDH, Ca2+-Mg2+-ATPase, and CAT enzymes in C. albicans.

Our results indicate that QAP reduces the activity of the SDH and Ca2+-Mg2+-ATPase in C. albicans. The SDH is located on the inner mitochondrial membrane, is vital for the TCA cycle and the electron transport chain (Cecchini, 2013), converting succinate to fumarate in an energy-dependent manner and producing FADH2, which transfers electrons to coenzyme Q (Rutter, Winge & Schiffman, 2010). Reduced SDH activity disrupts the normal function, affecting energy supply, gene regulation, and DNA replication, leading to inhibited cell growth or apoptosis. Ca2+-Mg2+-ATPase, found on the plasma and inner mitochondrial membrane, hydrolyzes ATP to release energy and extrudes calcium ions, maintaining intracellular calcium balance. A decrease in Ca2+-Mg2+-ATPase activity may disrupt intracellular calcium homeostasis and potentially lead to an imbalance in the electrochemical gradient across the membrane. Empirical evidence has demonstrated that elevated calcium concentrations can affect the TCA cycle, resulting in decreased SDH activity. Therefore, the reduction in SDH activity induced by QAP may be associated with alterations in calcium concentration due to decreased Ca2+-Mg2+-ATPase activity (Yang et al., 2014). However, further investigation is necessary to confirm whether there is an increase in cytoplasmic free calcium ions in C. albicans cells following QAP treatment. Catalase, a ubiquitous enzyme in C. albicans responsible for the elimination of reactive oxygen species (ROS) (Hamida et al., 2021), operates by catalyzing the decomposition of hydrogen peroxide into water and oxygen, thereby protecting the cell. Under normal conditions, ROS produced by cells are immediately decomposed by catalase. After treatment with QAP and FLC, there was an observed trend of increased CAT enzyme activity in C. albicans cells compared to the negative control group. This suggests that QAP treatment may induce ROS production in C. albicans, thereby activating endogenous ROS-detoxifying enzymes. Considering the cytotoxic properties of ROS, their excessive accumulation can lead to damage of cellular macromolecules.

To examine the effects of QAP on the morphology of C. albicans, this study utilized SEM to observe cellular alterations following treatment. The results indicated that, in comparison to the control group, C. albicans cells exposed to QAP exhibited significant cell shrinkage, with some cells displaying ruptured surfaces. Additionally, there was a marked reduction and flattening of budding sites. It is noteworthy that C. albicans is a dimorphic fungus that reproduces through budding. Notably, QAP effectively inhibited the growth of C. albicans hyphae. These observations are consistent with experimental results demonstrating the inhibitory effect of QAP on C. albicans growth, thereby confirming its efficacy in suppressing proliferation. Furthermore, these morphological alterations suggest that the inhibitory mechanism of QAP against C. albicans involves damage to cell walls and membranes, as well as inhibition of filamentous growth.

GO analysis indicated an enrichment of terms associated with cell wall organization, fungal-type cell wall biogenesis, and cation transport. This suggests that QAP may induce some degree of damage to the cell wall and membrane of C. albicans. It is plausible that QAP interacts with membrane proteins or translocate across membranes, potentially affecting processes such as metal ion transport and homeostasis, thereby impacting normal cellular metabolic activities in C. albicans cells.

The enrichment of the DNA replication proteins pathway in the KEGG database implies that the activity of QAP may induce cell death in a subset of C. albicans cells, thereby reducing DNA replication and resulting in the observed pathway enrichment. Furthermore, pathways related to energy metabolism, such as oxidative phosphorylation, acetate metabolism, and the TCA cycle, were significantly enriched. This observation suggests that the normal energy metabolism in C. albicans cells may be disrupted, consequently disturbing the balance between energy generation, supply, and utilization. Energy metabolism homeostasis is essential for the growth and proliferation of C. albicans, and its disruption can lead to starvation-induced cell death and substantial impairment of growth and reproductive functions. For instance, oxidative phosphorylation, which is localized in the inner mitochondrial membrane, constitutes the principal energy-generating mechanism for the cell. This process is responsible for approximately 90% of ATP production in C. albicans cells and is critical for fungal viability. Beyond providing the energy requisite for cellular survival and proliferation, cellular respiration plays a crucial role in synthetic metabolism and signal transduction pathways. Interference with this pathway can result in the accumulation of reactive oxygen species within cells, ultimately culminating in cell death.

The findings from the integrated analysis of GO, KEGG, and PPI networks suggest that SDH2 and ACO1 are pivotal proteins. SDH2, a protein located in the inner mitochondrial membrane, plays a crucial role in linking oxidative phosphorylation with electron transfer in C. albicans cells and serves as a marker enzyme indicative of mitochondrial function. Previous studies have demonstrated that the generation of the SDH2Δ/Δ mutant through gene knockout, coupled with virulence assays in various infection models, underscores the significance of SDH2 in virulence and hyphal formation. For instance, in the Caenorhabditis elegans model, infection with the wild-type strain resulted in more than 85% nematode mortality, whereas no mortality was observed with the SDH2Δ/Δ mutant. Similarly, in the murine model, the SDH2Δ/Δ mutant exhibited a significantly reduced survival rate and decreased fungal burden in the kidneys. Furthermore, the observation of hyphal growth in various hypha-inducing media and additional experiments confirmed that SDH2 modulates the electron transport chain and intracellular reactive oxygen species (ROS) levels (Bi et al., 2018), thereby playing a pivotal role in the virulence and hyphal formation of C. albicans. ACO1, a critical enzyme in the TCA cycle, is essential for cellular energy metabolism. Strijbis & Distel (2010) generated mutant strains with alterations in genes related to ACO1 and demonstrated, through growth analysis on various carbon source media, that ACO1 is crucial for the proliferation of C. albicans. A deficiency in ACO1 function results in significant growth defects. Consequently, QAP disrupts the energy metabolism of C. albicans’s by interfering with its normal respiratory processes (Strijbis & Distel, 2010).

The ALS4 gene predominantly encodes lectin-like sequence proteins that are situated on the cell wall surface of C. albicans, which are crucial for adhesion. Adhesion is a defining characteristic of mycelia and is a fundamental factor in the pathogenesis of C. albicans infections. According to the literature, inhibiting adhesion is essential for achieving antifungal effects (El-Houssaini et al., 2019). The significant down-regulation of ALS4 may suggest a defect in mycelial growth in C. albicans, resulting in a substantial impairment in its adhesion capability. Adhesion is critical in the initial stages of biofilm formation by C. albicans, implying that the cell membrane of C. albicans is similarly affected.

The PLB1 gene encodes phospholipase B, a virulence factor of C. albicans, which not only degrades the phospholipid components of host cells but also facilitates the synthesis and repair of fungal biofilms. Treatment with QAP leads to the downregulation of the PLB1 gene expression, ultimately resulting in the death of C. albicans.

The SIM1 gene encodes a protein with adhesin-like properties that is primarily involved in cell wall synthesis and morphological alterations. The cell wall is essential for maintaining cellular shape and plays a crucial role in enabling fungal cells to withstand external physical, chemical, and biological stressors, thus serving as a vital protective barrier. Significant variations in SIM1 expression suggest anomalies in the cell wall structure of C. albicans, indicating the necessity for upregulation of this gene to ensure cell wall integrity.

The ERG1 and MET6 genes are linked to the cell membrane, a critical organelle for sustaining the development, metabolism and maintenance of cellular homeostasis in C. albicans. As such, these genes are significant targets for pharmacological interventions aimed at affecting C. albicans cells. The fungal cell membrane of C. albicans is composed of phospholipid bilayers, sterols, and membrane proteins. Ergosterol, a vital component of fungal cell membranes, is crucial for maintaining membrane integrity (Jordá & Puig, 2020), permeability, and various other functions (Parks & Casey, 1995). The ERG1 gene encodes squalene monooxygenase, a pivotal enzyme in the biosynthesis of ergosterol. Meanwhile, the enzyme encoded by the MET6 gene is essential for thionine metabolic pathways, which are intricately connected to ergosterol synthesis by supplying precursor molecules necessary for this process and regulating various physiological functions, including cell membrane formation (Roje, 2006). Numerous studies have demonstrated that the MET6 gene, which encodes methionine synthase, is crucial for the development of C. albicans, with its absence hindering growth even in environments supplemented with exogenous methionine (Suliman, Appling & Robertus, 2007). Upon administration of QAP, there was a marked down-regulation in the expression levels of both ERG1 and MET6 genes, which may be attributed to cellular membrane damage or the inhibition of ergosterol biosynthesis pathways.

ZRT1 and PRA1 are genes associated with the transport of metal ions. The protein encoded by ZRT1 mediates the transmembrane transport of zinc ions in C. albicans. The PRA1p protein, encoded by PRA1, particularly in mycelial cells, possesses the ability to bind free zinc ions (Kurakado, Arai & Sugita, 2018) and facilitate their transfer to the ZRT1p membrane transporter located on the cell membrane surface. Subsequently, zinc ions are internalized for cellular utilization. The significant upregulation of these two genes further substantiates that treatment with QAP compromises the integrity of the cell membrane in C. albicans, inhibits hyphal growth, disrupts intracellular ion homeostasis, and necessitates the upregulation of related genes to sustain normal metabolic functions. QAP induces damage to the cell membrane of C. albicans, thereby affecting its permeability and disrupting its osmotic regulation capabilities (Carson, Mee & Riley, 2002; Lee & Kim, 2016).

The SDH2 gene encodes succinate dehydrogenase, an enzyme essential to aerobic respiration and the TCA cycle within cells, serving as a component of the mitochondrial respiratory chain. Studies have indicated that the absence of the SDH2 gene leads to impaired mycelial growth, a consequence attributed to the organism’s reliance on fermentable carbon sources, which subsequently disrupts cellular energy metabolism (Bi et al., 2018).

The CAT1 gene encodes catalase A, an enzyme crucial for regulating oxidative stress resistance in C. albicans cells. Treatment with QAP has been shown to significantly down-regulate CAT1 expression, suggesting that QAP may influence oxidative stress adaptability by down-regulating CAT1 expression. It is hypothesized that the disruption of the C. albicans cell membrane compromises its functionality, thereby increasing its susceptibility to external environmental factors (Wu et al., 2018).

The CST20 gene encodes a mitogen-activated protein kinase that plays a pivotal role as a positive regulator in the MAPK signaling pathway, modulating morphological changes in C. albicans and promoting filamentous growth within fungal colonies. Prior studies have shown that the deletion of CST20 inhibits mycelial formation, whereas treatment with QAP leads to the down-regulation of CST20 gene expression in C. albicans. Further validation has confirmed that QAP effectively suppresses mycelial growth and development.

PCK1, a gene crucial for C. albicans cellular metabolism, encodes the enzyme phosphoenolpyruvate carboxykinase, which is pivotal in the gluconeogenic pathway. This enzyme is essential for maintaining energy and carbon homeostasis under conditions of carbon limitation. The downregulation of PCK1 can lead to disruptions in the gluconeogenic process, resulting in an energy supply deficit and impaired adaptation to carbon deficient environments, potentially compromising the viability and fitness of C. albicans cell. Furthermore, the FBA1 gene encodes fructose-bisphosphate aldolase, an enzyme that plays a critical role in glycolysis, gluconeogenesis, and the pentose phosphate pathways (Wen et al., 2022). It has been reported that petroselinic acid can inhibit the mycelial formation of C. albicans by targeting fructose-1, 6-bisphosphate aldolase, thereby exerting an antifungal effect (Wang et al., 2024). In C. albicans cells treated with QAP, all three previously mentioned metabolic genes were significantly downregulated, indicating a disruption of the internal metabolic equilibrium and a profound impact on metabolic activities. Based on these changes in regulatory genes expression, a mechanistic diagram illustrating the inhibitory effect of QAP on C. albicans was developed (Fig. 17).

Figure 17 The mechanism of QAP components inhibiting C. albicans.

QAP may undermine the integrity of the cell wall and cell membrane of C. albicans to a certain extent. QAP may bind to membrane proteins or traverse the membrane to enter the cells, transmitting signals to the internal cell nucleus and causing a series of alterations in gene expression. The downregulation of FBA1 and CST20 expression suggests that the mycelial state of C. albicans is impacted, and its growth and development are inhibited by QAP. The downregulation of SIM1 and PLB1 expression and the upregulation of ALS4 expression indicate that the cell wall of C. albicans is compromised, with reduced adhesion and pathogenicity. The upregulation of ALS4 expression is in response to the damage caused by QAP to the biofilm. The downregulation of SDH2, ACO1, and PCK1 expression and the upregulation of CAT1 expression suggest that the normal operation of mitochondria is affected, potentially resulting in energy metabolism disorders. The downregulation of ERG1 and MET6 expression indicates that the synthesis process of ergosterol is hindered, further influencing the structure and function of the cell membrane. The upregulation of ZRT1 and PRA1 expression indicates that the transport and homeostasis of metal ions inside and outside the cells of C. albicans are disrupted, further suggesting a decline in the cells’ ability to uptake related metal ions. This may be due to the damage to ion transport proteins and ion channels on the cell membrane or cell wall, and the upregulation of related genes is necessary to stabilize cell metabolism. Blue arrows indicate that metal ions/QAP enter the cells of C. albicans through channel proteins/membrane proteins. Black arrows represent the sites of action of relevant genes within the cells of C. albicans. Red and yellow arrows are employed to signify increased and decreased gene expression, respectively.

In conclusion, QAP exhibits substantial antifungal activity against C. albicans by targeting multiple cellular sites. These mechanisms include inducing damage to the cell wall and membrane, inhibiting hyphal growth, and disrupting normal cellular respiration processes such as oxidative phosphorylation, which consequently leads to dysregulation of energy metabolism in C. albicans cells. However, a limitation of this study is the inability to conduct more in-depth research on the hyphae of C. albicans, which constitute its primary virulence factor.

The QAP exhibits considerable potential as a natural therapeutic agent devoid of side effects within the framework of traditional Chinese medicine for the treatment of C. albicans infections. This discovery presents a promising opportunity to leverage quinoa resources in the pharmaceutical industry, thereby augmenting its economic value. Future research should focus on further exploring the nutritional and medicinal properties of quinoa, which may contribute to the management of various chronic diseases and the promotion of human health and development (Hussain et al., 2021).

The diversity of protein types in the crude extract of QAP is notably high (Fig. 6), posing challenges for its separation and purification. Future efforts will involve the application of various separation and purification techniques, such as anion/cation exchange, hydrophobic chromatography, and ultrafiltration to isolate and purify the existing 18 quinoa proteins. The goal is to narrow down the range of active quinoa proteins or to isolate a single active protein component that capable of inhibiting C. albicans.

Conclusions

The findings of this study present an innovative methodology for the isolation and purification of QAP, which exhibits inhibitory effects against C. albicans. This advancement provides new insights into the development of natural pharmaceuticals devoid of side effects. In this investigation, QAP was preliminarily identified as an 11S seed storage protein using ammonium sulfate precipitation, Sephadex gel chromatography, and proteomic data analysis. Further analyses, including the assessment of SDH, Ca2+-Mg2+-ATPase, and CAT activity, SEM results, and transcriptome sequencing data, reveal that the QAP induces cellular damage to both the cell wall and membrane of C. albicans cells, inhibits hyphal growth, modulates membrane permeability, and disrupts molecular pathways including ergosterol synthesis, cellular metabolism, metal ion transport, and homeostasis. These findings elucidate the mechanistic pathways through which the QAP exerts its inhibitory effects on C. albicans.

Consequently, these research findings provide substantial theoretical support for the development of multi-target natural therapeutics that effectively inhibit C. albicans and reduce the likelihood of drug resistance. This study has successfully narrowed the range of QAP to 18 types. In future research, isoelectric focusing electrophoresis will be employed to identify QAP. This technique leverages the differences in the isoelectric points of proteins, allowing them to migrate to their respective isoelectric points within an electric field, thereby facilitating separation. The method’s high resolution enables precise differentiation of proteins with similar isoelectric points. Through this approach, it is anticipated that single active proteins can be isolated from the current mixed proteins, their characteristics and antifungal mechanisms elucidated, and new directions for the research into the antifungal properties of QAP explored.

Supplemental Information

Text S1 Operating Procedures of ELISA Experiment and Standard Curve

Data S1 Effect Diagrams of MIC of QAP and FLC

Table S1 List of specific primer sequences

Table S2 GO terms

Table S3 KEGG pathway

Table S4 PPI analysis results

Supplemental Information 7 Raw data for Figure 2

Supplemental Information 8 Raw data for Figure 4

Supplemental Information 9 Raw data for Figure 6A

Supplemental Information 10 Raw data for Figure 6B

Supplemental Information 11 Raw data for Figure 7

Supplemental Information 12 Raw Data for Figure 8

Supplemental Information 13 Raw data for Figure 9

Supplemental Information 14 Raw data for Figure 10

Supplemental Information 15 Raw data for Figure 11

Supplemental Information 16 Raw data for Figure 14F

Supplemental Information 17 Raw data for Figure 16

Supplemental Information 18 MIQE Checklist

Supplemental Information 19 The detailed procedure for the qPCR experiment

We thank Mingjie Chen (Shanghai NewCore Biotechnology Co., Ltd.) for providing data analysis and visualization support. DNA, Mitochondrion, Ion Channel cartoon images used in figures were obtained from Scidraw.io.

Additional Information and Declarations

Competing Interests

Author Contributions

Data Availability

The authors declare there are no competing interests.

Xufei Zhang conceived and designed the experiments, performed the experiments, analyzed the data, prepared figures and/or tables, authored or reviewed drafts of the article, and approved the final draft.

Chunmei Zheng performed the experiments, analyzed the data, prepared figures and/or tables, and approved the final draft.

Wenxuan Ge performed the experiments, analyzed the data, prepared figures and/or tables, and approved the final draft.

Xueying Li performed the experiments, analyzed the data, prepared figures and/or tables, and approved the final draft.

Xiuzhang Wang performed the experiments, analyzed the data, prepared figures and/or tables, and approved the final draft.

Yanxia Sun conceived and designed the experiments, performed the experiments, analyzed the data, prepared figures and/or tables, authored or reviewed drafts of the article, and approved the final draft.

Xiaoyong Wu conceived and designed the experiments, performed the experiments, analyzed the data, prepared figures and/or tables, authored or reviewed drafts of the article, and approved the final draft.

The following information was supplied regarding data availability:

The data is available at NCBI SRA: PRJNA1166727.

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
