# Peer review of "Preparation and analysis of quinoa active protein (QAP) and its mechanism of inhibiting Candida albicans from a transcriptome perspective"

_PeerJ, doi:10.7717/peerj.18961_

## Round 0.1 · original submission · Major Revisions

As noted by the reviewers, the manuscript requires modifications for improved clarity. Therefore, the authors should thoroughly address all queries raised by the reviewers.

Reviewer 1 ·

Basic reporting

(1)The paper's title is somewhat misleading. Indeed, the study employed proteomic techniques to investigate the composition of quinoa's active proteins, and RNA sequencing was utilized to assess the impact of Quinoa Active Peptides (QAP) on the transcriptome of Candida albicans. It would be advisable for the authors to reconsider and suggest a title that more accurately reflects the research's focus.
(2)In the Introduction section, the author should elaborate on the advantages of the QAP component analysis method employed in this study compared to previous research.
(3)In the Introduction section, the author should elaborate on why QAP potentially possesses anti-C. albicans activity.
(4)There are cases of omitted citations in the experimental result images, for instance, Figure 7c.
(5)It is recommended to consolidate some of the experimental content into a single, coherent figure, rather than creating a single figure that encompasses multiple, disparate contents and is subsequently referenced by numerous experimental results, as exemplified by Figure 7.
(6)Figure 3 lacks a scale bar.
(7)In Figure 7, could you elaborate on how the radius of the inhibition zone is defined? Specifically, is it measured as the radius of the circle formed by the filter paper disk, or as the width of the ring that forms outside the perimeter of the filter paper disk? According to the data presented in Figure 7B, the radius of the inhibition zone is defined as the radius of the circle traced by the filter paper disk. Nonetheless, this particular definition appears to be flawed, as it suggests that even a saline solution devoid of antifungal activity would possess an inhibition zone and an associated MIC value.
(8)Line 348-354. Based on the content depicted in Figure 9, this study did not investigate the morphological transition of C. albicans but solely focused on alterations occurring on the surface of its cells. It is recommended that the authors augment their research with experiments that assess the inhibition of hyphal formation by QAP in C. albicans, and adjust the outcomes of this segment to reflect the impact of QAP on the surface of C. albicans cells.
(9)The legend for Figure 2 should offer clear information to aid readers in interpretation, specifying what elements a, b, c, and d in the figure denote.
(10)The dyeing method of Figure 5 should be explained.
(11)What does "CK" signify in Figure 7? A note should be included in the figure legends to clarify this.
(12)The clarity of Figure 10 is inadequate; the numbers within the figure are indistinct. It is advisable to enhance the clarity of Figure 10 to improve its readability.
(13)Figure 13 is challenging to interpret due to the absence of annotations for numerous elements within the diagram, including the arrows of various colors and the red circles.
(14)Figure 4A was not cited appropriately in the manuscript.

Experimental design

(1)Lines 167-169. The definition of MIC should be unmistakably clear. What exactly is the minimum diameter of the drug concentration that is designated as the MIC?
(2) Furthermore, it is advisable to utilize the microdilution method for determining the MIC values of various compounds against C. albicans.
(3)Lines 170-177. In the "Growth kinetics of C. albicans under QAP" methods section, it is essential to clearly define key parameters of the experiment, including the strains utilized, the concentration of the fungal solution, the concentration of QAP, and the instruments and equipment employed to measure absorbance values.
(4)Lines 188-192. The methodology section outlined in "Activity analysis of SDH, Ca2+-Mg2+-ATPase, and CAT" lacks specificity, posing challenges for researchers aiming to replicate the experiment. It is imperative that the following details be explicitly provided: the precise protein concentrations utilized for assessing the activity of these three enzymes, the identities and concentrations of the substrates involved, the temperature under which the experiments were performed, the duration of the reactions, and the procedures employed for data processing.
(5)Line 195. In which medium is the hyphal induction of C. albicans carried out?
(6)Lines 325-327. I am confused about how the MIC value of QAP against C. albicans was measured as 182 μg/mL using the filter paper disc diffusion method. Shouldn't the results obtained by the disc diffusion method be in units of length, such as millimeters or centimeters?
(7)Lines 328-354. Why is a high concentration of 0.78 mg/mL of QAP selected for subsequent phenotypic experiments, despite the MIC of QAP against C. albicans being 182 μg/mL?

Validity of the findings

(1)Lines 325-327. The author is tasked with interpreting the results and conducting a detailed statistical analysis, including the provision of the P-value. The existing description is overly vague, leaving the reader uncertain about the specific MIC of the compound and whether the observed difference is statistically significant compared to control.
(2)Line 309. The author claims that "ammonium sulfate fractionation is an effective method for isolating QAP with inhibitory properties against C. albicans" lacks any foundational support. The ammonium sulfate method merely elevated the concentration of proteins within the 18.4 to 25.0 kDa molecular weight range; however, it remains uncertain whether these proteins exhibit anti-C. albicans activity
(3)Lines 328-331. The author's depiction is perplexing. Precisely what does AKP stand for? Why is it essential to gauge the activity of AKP? The AKP activity within the QAP treatment cohort measures 0.62 ± 0.09 U/mL, while the control group's AKP activity is also recorded at 0.62 ± 0.09 U/mL. Consequently, it is unclear why it is asserted that the QAP treatment group exhibits higher AKP activity compared to the control group.
(4)Lines 341-347. The unit of enzyme activity ought not to be articulated in terms of mass concentration (mg/mL).
(5)Lines 341-347. The data presentation lacks standardization; it should be formatted as mean ± error.
(6)Line 372-393.The outcomes of RNA sequencing should be confirmed through phenotypic validation. It is not sufficient to deduce the antifungal mechanism of QAP solely from the quantity of genes exhibiting altered expression levels. This is because such variations could potentially be a consequence of QAP's inhibition of C. albicans growth, rather than the underlying cause of the inhibition.
(7)Line 391. Which biological processes are SDH2 and ACO1 essential proteins for? How were their crucial roles ascertained? These questions should be clearly elaborated.
(8)Line 392. The description of the qRT-PCR experimental results should be more detailed, including specific information on which genes were detected and the observed fold changes.

Additional comments

(1)“C. albicans”ought to be displayed in italics, rather than in a regular font. This non-conforming format can be found on lines 25, 28, 32, and 35. Furthermore, there are numerous other occurrences throughout the text where the author needs to make thorough corrections.
(2)Line 63. "Quinoa protein" — is it a protein complex or a pure, isolated protein? If it constitutes a protein complex, it ought to be articulated in its plural form. Conversely, if it is a pure protein, then there is no necessity for a component analysis. The author is requested to furnish suitable clarification regarding this matter.
(3)The author alternates between the terms "quinoa" (e.g., line 67) and "quinoa seeds" (e.g., line 70), which can be perplexing. It would be beneficial for the author to delineate the distinctions and correlations between these two terms to offer readers a more transparent comprehension of the article's subject matter.
(4)Line 75. C. albicans is a fungus, not a bacterium.
(5)Lines 80-82. The annual incidence of C. albicans infections and the mortality rate associated with these infections lack precision. It is essential to reference the latest epidemiological data.
(6)Line 118. The numeral and its corresponding unit in "4 ℃" ought to be separated by a space. Throughout the article, there are several similar instances that require the author's meticulous and comprehensive correction.
(7)Line 171. The term ‘bacterial solution’ ought to be replaced with ‘fungal solution’. The article contains multiple instances where the author incorrectly identifies C. albicans as bacteria, necessitating thorough revisions.
(8)Line 188. The full forms of "SDH" and "CAT" should be provided.
(9)Line 307. Appropriate literature should be cited to indicate which previous studies are confirmed.
(10)Line 338. Be mindful of maintaining consistency in the significant figures for the numerical values "1.40 ± 0.008" and "0.621 ± 0.01".

Reviewer 2 ·

Basic reporting

1. Mark the length of ZOI in fig 3.
2. Explain the fig 4 in its legend clearly and detailed.
3. What is the reason of too many protein band in lane 8 of fig 5.
4. The diameter of ZOI is missing in fig 8 A.
5. Show the progressive flattening of budding sites in fig 9 with arrow.
6. Data of Table 1 represents what? Mention it in a row.
7. Thoroughly check the typos error throughout the MS. For eg Candida. Albicans should be Candida albicans.
8. In lines 75-76 of introduction section, please remove the old references like Desai et al. 2015 and Kitahara et al. 2015. In place of this add recent references from the year 2024 DOI:10.1016/j.genrep.2024.102044 and year 2023 doi: 10.1089/mdr.2022.0347.

Experimental design

1. How did you classify QAP protein
2. Write the method for generation of Heatmap in methodology section clearly.

Validity of the findings

Conclusion must explain future direction of the work.

---

## Round 0.2 · accepted · Accept

Since the authors have satisfactorily addressed the reviewers' queries, the manuscript is suitable for acceptance and publication.

Reviewer 1 ·

Basic reporting

None

Experimental design

None

Validity of the findings

None

Additional comments

None

Reviewer 2 ·

Basic reporting

In the revised version of the manuscript, the authors have adequately addressed the concerns raised. The substantial revision of the manuscript makes it appropriate for acceptance. I recommend its acceptance for publication.

Experimental design

In the revised version of the manuscript, the authors have adequately addressed the concerns raised. The substantial revision of the manuscript makes it appropriate for acceptance. I recommend its acceptance for publication.

Validity of the findings

In the revised version of the manuscript, the authors have adequately addressed the concerns raised. The substantial revision of the manuscript makes it appropriate for acceptance. I recommend its acceptance for publication.

Additional comments

In the revised version of the manuscript, the authors have adequately addressed the concerns raised. The substantial revision of the manuscript makes it appropriate for acceptance. I recommend its acceptance for publication.